# Impact of Annual Change in Geriatric Nutritional Risk Index on Mortality in Patients Undergoing Hemodialysis

**DOI:** 10.3390/nu12113333

**Published:** 2020-10-29

**Authors:** Takahiro Yajima, Kumiko Yajima, Hiroshi Takahashi

**Affiliations:** 1Department of Nephrology, Matsunami General Hospital, Gifu 501-6062, Japan; 2Department of Internal Medicine, Matsunami General Hospital, Gifu 501-6062, Japan; green_tea_1324@yahoo.co.jp; 3Division of Medical Statistics, Fujita Health University School of Medicine, Aichi 470-1192, Japan; hirotaka@fujita-hu.ac.jp

**Keywords:** hemodialysis, geriatric nutritional risk index (GNRI), annual change in GNRI (ΔGNRI), all-cause mortality, cardiovascular mortality

## Abstract

Regular nutritional assessment may decrease the mortality rate in patients undergoing hemodialysis. This study aimed to evaluate whether annual change in geriatric nutritional risk index (ΔGNRI) can precisely predict mortality. We retrospectively examined 229 patients undergoing hemodialysis who measured geriatric nutritional risk index (GNRI). Patients were divided into four groups according to the baseline GNRI of 91.2, previously reported cutoff value, and declined or maintained GNRI during the first year (ΔGNRI < 0% vs. ΔGNRI ≥ 0%): Group 1 (G1), GNRI ≥ 91.2 and ΔGNRI ≥ 0%; G2, GNRI ≥ 91.2 and ΔGNRI < 0%; G3, GNRI < 91.2 and ΔGNRI ≥ 0%; and G4, GNRI < 91.2 and ΔGNRI < 0%. They were followed for mortality. During a median follow-up of 3.7 (1.9–6.9) years, 74 patients died, of which 35 had cardiovascular-specific causes. The GNRI significantly decreased from 94.8 ± 6.3 to 94.1 ± 6.7 in the first year (*p* = 0.035). ΔGNRI was negatively associated with baseline GNRI (ρ = −0.199, *p* = 0.0051). The baseline GNRI < 91.2 and ΔGNRI < 0% were independently associated with all-cause mortality (adjusted hazard ratio (aHR) 2.59, 95%, confidence interval (CI) 1.54–4.33, and aHR 2.33, 95% CI 1.32–4.32, respectively). The 10-year survival rates were 69.8%, 43.2%, 39.9%, and 19.2% in G1, G2, G3, and G4, respectively (*p* < 0.0001). The aHR value for G4 vs. G1 was 3.88 (95% CI 1.62–9.48). With regards to model discrimination, adding ΔGNRI to the baseline risk model including the baseline GNRI significantly improved the net reclassification improvement by 0.525 (*p* = 0.0005). With similar results obtained for cardiovascular mortality. We concluded that the ΔGNRI could not only predict all-cause and cardiovascular mortality but also improve predictability for mortality; therefore, GNRI might be proposed to be serially evaluated.

## 1. Introduction

Malnutrition is common and associated with increased risks of mortality in patients undergoing hemodialysis (HD) [1,2]. Regular nutritional assessment is recommended to reduce mortality in patients undergoing HD [3]. Moreover, protein-energy wasting (PEW) frequently develops in this population [4].

As for the screening methods for nutritional status, the malnutrition inflammation score (MIS) is the most validated method for screening patients undergoing HD with nutritional risks compared with other tools [3,5,6]. However, the MIS needs subjective assessment by well-trained examiners to obtain consistent results. The geriatric nutritional risk index (GNRI) was an objective and simple method based on serum albumin levels, height, and weight to evaluate nutritional status [7,8]. Some previous studies showed that the GNRI is an effective tool for stratifying risks of malnutrition and identifying nutrition-related risks of mortality or cardiovascular events in patients undergoing HD [9,10,11,12]. A meta-analysis conducted by Xiong et al. concluded that GNRI is a strong predictor for both cardiovascular and all-cause mortality in HD patients [13]. The nutritional status may constantly change; therefore, the prognostic value of GNRI for adverse clinical outcomes might also change over time. Recently, Lee et al. reported that changes in GNRI were significantly related to increased risks of cardiovascular events in patients undergoing incident peritoneal dialysis [14]. However, the relationships between changes in GNRI and mortality remain unclear in HD patients.

The present study aimed to investigate the associations between the annual change in GNRI (ΔGNRI) and all-cause and cardiovascular mortality in patients undergoing maintenance HD. Moreover, we examined whether the ΔGNRI can improve the predictability of mortality when it was added to the established risk factors including the baseline GNRI.

## 2. Materials and Methods

### 2.1. Study Participants

We carried out a retrospective cohort study involving patients who underwent maintenance HD therapy for at least 6 months. This study was performed by perusing medical records of the outpatient clinic of Matsunami General Hospital (Kasamatsu, Gifu, Japan) from January 2008 to December 2019. We screened 229 patients undergoing maintenance HD at baseline. Patients’ data were completely anonymized before access, and the requirement for informed consent was waived. The present study adhered to the principles of the Declaration of Helsinki and was approved by the ethics committee of Matsunami General Hospital (No. 469).

### 2.2. Data Collection

The following data of the patients were collected using the medical records: age; sex; underlying kidney disease; HD duration; history of alcohol, smoking, hypertension, diabetes, and cardiovascular disease (CVD); height; and dry weight. In the present study, CVD was defined as myocardial infarction, angina pectoris, heart failure, stroke, and peripheral artery disease. Diabetes was defined as a history of diabetes or use of glucose lowering agents. Hypertension was defined as systolic blood pressure ≥ 140 mmHg or diastolic blood pressure ≥ 90 mmHg before HD or use of antihypertensive drugs. Blood samples were obtained in the supine position prior to HD sessions, which were conducted on either a Monday or Tuesday. GNRI was calculated using the following formula: GNRI = (14.89 × albumin (g/dL)) + (41.7 × (dry weight/ideal body weight)) [8]. When the dry weight surpassed the ideal body weight, the element of “(dry weight/ideal body weight)” was set to 1. The GNRI was calculated at enrollment point and after one year; thereafter, ΔGNRI was calculated by subtracting the baseline GNRI from GNRI after one year.

### 2.3. Follow-Up Study

The primary endpoint was all-cause mortality. The secondary endpoint was cardiovascular mortality. Patients were divided into four groups according to the baseline GNRI of 91.2, a previously reported cutoff value, and declined or maintained GNRI in the first year: Group 1 (G1), GNRI ≥ 91.2 and ΔGNRI ≥ 0%; G2, GNRI ≥ 91.2 and ΔGNRI < 0%; G3, GNRI < 91.2 and ΔGNRI ≥ 0%; and G4, GNRI < 91.2 and ΔGNRI < 0%. These patients were followed until December 2019.

### 2.4. Statistical Analysis

Normally distributed variables were expressed using mean ± standard deviation, whereas non-normally distributed variables were expressed as median and interquartile range. The differences among the four subgroups divided by each cutoff value for the baseline GNRI and the ΔGNRI were compared using one-way analysis of variance or Kruskal–Wallis test for continuous variables and the chi-squared test for categorical variables. The association between the ΔGNRI and baseline GNRI was evaluated using Spearman’s rank correlation coefficient. The survival rate was estimated using the Kaplan–Meier method, and the difference was analyzed using the log-rank test. Hazard ratios (HRs) and 95% confidence intervals (CIs) for mortality were calculated by Cox regression analysis. The multiple regression model included all covariates that were significant at a *p*-value of <0.05 in the univariate analysis.

The C-index, net reclassification improvement (NRI), and integrated discrimination improvement (IDI) were used to assess whether predictive accuracy of mortality could improve after the addition of ΔGNRI to the baseline model, including the baseline GNRI. The C-index was defined as the area under the receiver operating characteristic curve between individual predictive probabilities for mortality and incidence of mortality [15]. It was compared between the baseline model with all established risk factors including the baseline GNRI and the enriched model adding the ΔGNRI. The NRI was defined as a relative indicator of the number of patients with improved predicted mortality risk, and the IDI was defined as an average improvement in predicted mortality risk after the addition of new variables to the baseline model [16]. Statistical analyses were conducted with IBM SPSS version 21 (IBM Corp., Armonk, NY, USA). A *p*-value of <0.05 was considered statistically significant.

## 3. Results

### 3.1. Baseline Characteristics

Patients’ baseline characteristics are summarized in Table 1. The mean age was 63.6 ± 13.8 years, and 69.4% were men. The median HD duration was 0.53 (0.51–3.80) years, with diabetes (45.8%) and history of CVD (62.4%). The baseline GNRI was 94.0 ± 7.0.

### 3.2. Associations of Baseline GNRI with Mortality

A median follow-up period of 3.7 (1.9–6.9) years showed that 74 patients died due to CVD (35 (47.3%), heart failure, 13; sudden death, 11; stroke, 7; and myocardial infarction, 4), infection (19 (25.7%)), malignancy (11 (14.9%)), and other causes (9 (12.2%)) (Figure 1).

In the multivariate Cox proportional hazards analysis adjusted for age, creatinine level, and C-reactive protein level, which were significant at *p*-value < 0.05 in the univariate analysis, the baseline GNRI was a significant predictor for all-cause mortality (adjusted hazard ratio (aHR), 0.94; 95% confidence interval (CI), 0.91–0.98; *p* = 0.0033). We divided patients by cutoff value of GNRI 91.2 into low and high groups (GNRI < 91.2 vs. GNRI ≥ 91.2). The 5-year all-cause survival rates were 82.1% and 43.9%, respectively (*p* < 0.0001). The 11-year all-cause survival rates were 53.7% and 22.0%, respectively (*p* < 0.0001) (Figure 2a). The lower baseline GNRI was independently associated with increased risks of all-cause mortality (aHR, 2.59; 95% CI, 1.54–4.33; *p* = 0.0004) (Table 2). Similar results were obtained for cardiovascular mortality (Figure 2d, Table 2).

### 3.3. Associations of ΔGNRI and Baseline GNRI with Mortality

Within the first follow-up year, 20 patients died (cardiovascular-specific cause, 7; heart failure, 4; stroke, 2; myocardial infarction, 1; infection, 5; malignancy, 5; others, 3), 8 patients transferred to another HD unit, and two patients did not reach one year follow-up. These 30 patients were excluded, and finally, 199 patients were analyzed for associations of the ΔGNRI and baseline GNRI with mortality (Table 1). The GNRI significantly decreased from 94.8 ± 6.3 to 94.1 ± 6.7 in the first year (*p* = 0.035), and the ΔGNRI was −0.4% (−3.6 to 2.2%). The decline of annual change in GNRI occurred in 78 patients (52.3%) of higher GNRI group (GNRI ≥ 91.2) and 24 patients (48.0%) of lower GNRI group (GNRI < 91.2), respectively. ΔGNRI was negatively associated with baseline GNRI (ρ = −0.199, *p* = 0.0051).

In the multivariate Cox proportional hazards analysis, the ΔGNRI was a significant predictor for all-cause mortality (aHR, 0.89; 95% CI, 0.83–0.95; *p* = 0.0006) (Table 2). Patients were divided by the declined or maintained GNRI in the first year (ΔGNRI < 0% vs. ΔGNRI ≥ 0%). The 4-year all-cause survival rates were 70.5% and 88.3%, respectively (*p* = 0.013). The 10-year all-cause survival rates were 41.3% and 63.7%, respectively (*p* = 0.016) (Figure 2b). The declined ΔGNRI was an independent predictor for all-cause mortality (aHR, 2.33; 95% CI, 1.32–4.32; *p* = 0.0032) (Table 2). Moreover, patients were divided by each cutoff value of the baseline GNRI and ΔGNRI into G1, G2, G3, and G4 groups. The 4-year all-cause survival rates were 91.8%, 79.0%, 79.8%, and 38.4%, in G1, G2, G3, and G4, respectively (*p* < 0.0001). The 10-year all-cause survival rates were 69.8%, 46.8%, 39.9%, and 19.2%, in G1, G2, G3, and G4, respectively (*p* < 0.0001) (Figure 2c). Moreover, aHRs for all-cause mortality were obtained as follows: 2.68 (95% CI 1.31–5.93, *p* = 0.0061) for G2 vs. G1, 2.57 (95% CI 0.84–7.27, *p* = 0.095) for G3 vs. G1, and 3.88 (95% CI 1.62–9.48, *p* = 0.0026) for G4 vs. G1 (Table 2). Similar results were obtained for cardiovascular mortality (Figure 2e,f, Table 2).

### 3.4. Model Discrimination 

The C-index for all-cause mortality increased from 0.702 to 0.733 with the addition of ΔGNRI to the established risk model including age, creatinine level, C-reactive protein level, and baseline GNRI (*p* = 0.39), but it did not reach statistical significance. However, the NRI and IDI for all-cause mortality significantly improved by the addition of the ΔGNRI to each established risk model, even including baseline GNRI (0.525 (*p* = 0.0005) and 0.055 (*p* = 0.0005), respectively) (Table 3). Similar results were obtained for cardiovascular mortality (Table 3).

## 4. Discussion

The main findings of the present study demonstrated that the ΔGNRI was significantly negatively correlated with the baseline GNRI and could predict all-cause and cardiovascular mortality in maintenance HD patients. Moreover, the predictive accuracy of mortality was improved after adding ΔGNRI to a model with established risk factors, including the baseline GNRI. These findings suggest that regular nutritional assessment using the GNRI may be useful to more accurately predict mortality in this population.

PEW, which is a state of malnutrition defined by a loss of muscle and fat in the presence of chronic inflammation, is prevalent and associated with increased risks of cardiovascular and all-cause mortality in HD patients [4,17,18]. Kalanter-Zadeh et al. proposed the MIS as a gold standard nutritional screening tool in evaluating malnutrition of patients undergoing HD [3]. The MIS was associated with morbidity and mortality in HD patients, but it requires a subjective assessment [3,19]. In contrast, the GNRI is used as a simple objective nutritional screening tool in this population [8,19]. According to Yamada et al., GNRI was negatively correlated with the MIS [8]. When malnutrition was defined as MIS ≥ 6, the cutoff value of GNRI was <91.2 [8]; we used this value in the analysis. The GNRI can be used in the assessment of PEW in patients undergoing HD [20]. Moreover, many studies have reported that the GNRI is a strong predictor for cardiovascular and all-cause mortality [9,10,11,12,13]. Although the predictability for mortality of the MIS was better than the GNRI [6,19], Chen et al. have recently shown in the largest cohort that the predictability for cardiovascular and all-cause mortality of the GNRI was similar to those of the MIS [21]. Furthermore, the GNRI can be more easily calculated compared to the MIS; therefore, the GNRI may be useful for repeated nutritional assessments.

There are only a few studies evaluating the relationships between GNRI changes and various clinical outcomes in HD patients. Beberashvili et al. reported that the changes in GNRI were associated with the changes in nutritional biomarkers, body composition parameters, and interleukin-6 level in patients undergoing HD [19]. Thus, the GNRI may be a useful tool for longitudinal assessment of nutritional status in this population. In contrast, Lee et al. recently reported that the change in GNRI was significantly correlated with the risks of cardiovascular events in patients undergoing incident peritoneal dialysis [14]. However, the associations between longitudinal changes in GNRI and mortality have never been studied.

In the present study, all-cause and cardiovascular survival rates were lower in the lower baseline GNRI group and the declined GNRI group, respectively. Furthermore, all-cause and cardiovascular survival rates were lowest in the lower baseline GNRI and the declined GNRI group (G4). Because a median follow-up period was short, we also calculated survival rates for shorter intervals of time. However, these results were similar to those of longer intervals of time. Thus, we think that many events’ data were missing if we had shortened the follow-up period, therefore we showed longer follow-up data.

In this study, the lower baseline GNRI and declined GNRI in the first year were independently associated with increased risks of all-cause and cardiovascular mortality in patients undergoing maintenance HD, respectively. Patients with lower baseline GNRI and declined GNRI (G4) had the highest risks of mortality. The GNRI significantly decreased in one year, and ΔGNRI was negatively associated with the baseline GNRI. Moreover, patients with higher baseline GNRI and declined GNRI (G2) had independently higher risks of mortality compared with patients with higher baseline GNRI and maintained GNRI (G1), but patients with lower baseline GNRI and maintained GNRI (G3) did not have independently increased risks of mortality compared with G1. These results demonstrated that patients with good baseline nutritional status may have increased risks of mortality if thereafter their nutritional status deteriorates. Therefore, repeated nutritional assessments may be useful in stratifying the risks for mortality. 

As for the model discrimination, the C-index for all-cause mortality increased by the addition of ΔGNRI to the established risk model including baseline GNRI from 0.702 to 0.733 (*p* = 0.39), but it did not reach statistical significance. This might be because of the small number of patients. However, the addition of ΔGNRI to the predicting model significantly improved the NRI (0.525, *p* = 0.0005) and IDI (0.055, *p* = 0.0005). The NRI relatively indicates how many patients improve their predicted probabilities, and the IDI also represents the average improvement in predicted probabilities, respectively [16]. In other words, 52.5% of patients had improved the predictability and the average improved predicted probabilities increased by 0.055 when ΔGNRI was added to a predicting model including baseline GNRI. Moreover, similar results were achieved for cardiovascular mortality. However, recently, some statisticians raised a concern regarding the overestimation of the improvement of predictability among predicting models using the NRI [22,23]. Although the NRI has been used to discriminate predicting models among numerous studies, this point should be heeded, and our results might have to be confirmed using other evaluation tools. Thus, the present study suggested that the predictability for all-cause and cardiovascular mortality might be improved after the addition of ΔGNRI to a model with established risk factors, even including the baseline GNRI. These findings might also support the clinical usefulness of regular nutritional assessments using the GNRI to predict mortality in patients undergoing maintenance HD.

Several limitations should be considered in this study. First, the present retrospective study was based on a small number of patients undergoing maintenance HD in a single center. The 10-year survival data might be unstable because of short median follow-up period. Second, this study only included Japanese patients undergoing HD, our findings were too limited to be generalized to patients undergoing HD in other countries. Third, we evaluated the relationships between the annual change in GNRI and mortality, but the optimal duration of changes in GNRI remains unknown. Fourth, the results of model discrimination were controversial. The NRI may potentially overestimate the improvement of predictability. Our results might have to be confirmed using other evaluation tools. Therefore, a further large-scale multicenter study may be needed to validate our results. Nevertheless, the present study provided the first associations between the annual change in GNRI and mortality.

## 5. Conclusions

The annual change in GNRI was negatively associated with the baseline GNRI and accurately predicted all-cause and cardiovascular mortality in patients undergoing maintenance HD. Moreover, the predictive accuracy of mortality might be improved after adding the annual change in GNRI into a model with established risk factors, including the baseline GNRI. Therefore, the GNRI might be proposed to be serially evaluated to precisely predict mortality in this population.

## Figures and Tables

**Figure 1 nutrients-12-03333-f001:**
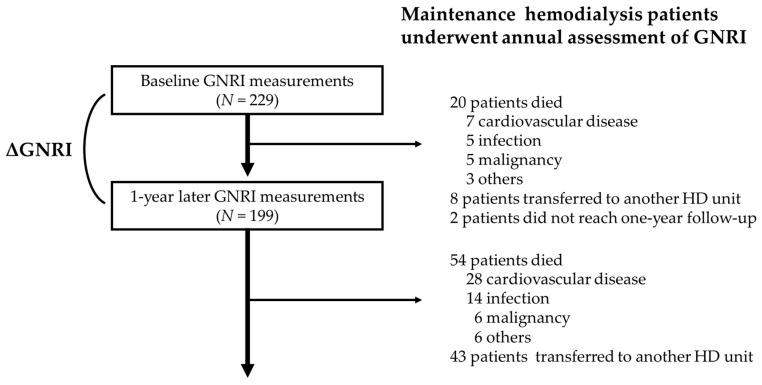
Flow diagram of the present study. GNRI, geriatric nutritional risk index; ΔGNRI, annual change in GNRI; HD, hemodialysis.

**Figure 2 nutrients-12-03333-f002:**
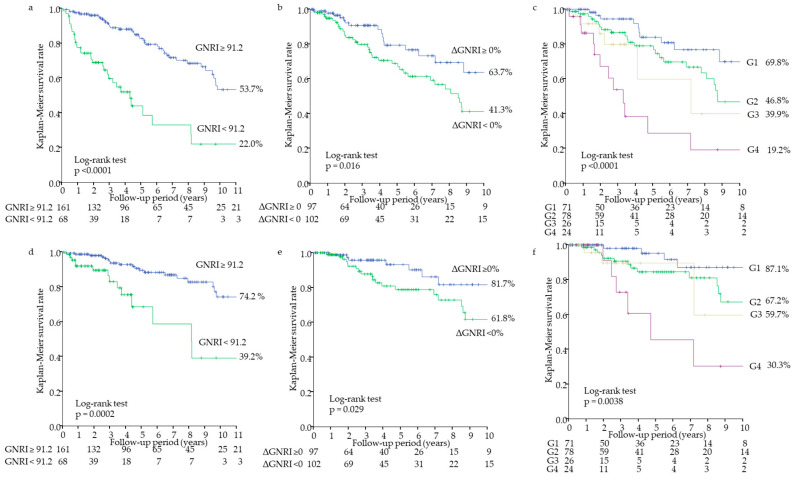
Kaplan–Meier survival curves for all-cause and cardiovascular mortality. All-cause mortality for GNRI < 91.2 vs. GNRI ≥ 91.2 (**a**), ΔGNRI < 0% vs. ΔGNRI ≥ 0% (**b**), and among the four groups divided by the GNRI and ΔGNRI (**c**). Cardiovascular mortality for GNRI < 91.2 vs. GNRI ≥ 91.2 (**d**), ΔGNRI < 0% vs. ΔGNRI ≥ 0% (**e**), and among the four groups divided by the GNRI and the ΔGNRI (**f**). G1 (group 1), GNRI ≥ 91.2 and ΔGNRI ≥ 0%; G2, GNRI ≥ 91.2 and ΔGNRI < 0%; G3, GNRI < 91.2 and ΔGNRI ≥ 0%; and G4, GNRI < 91.2 and ΔGNRI < 0%. GNRI, geriatric nutritional risk index; ΔGNRI, annual change in GNRI.

**Table 1 nutrients-12-03333-t001:** Baseline characteristics of the study patients.

Variables	All Patients	Four Groups (*N* = 199)				
	Baseline (*N* = 229)	G1 (*N* = 71)	G2 (*N* = 78)	G3 (*N* = 26)	G4 (*N* = 24)	*p*-Value
Age (years)	63.6 ± 13.8	60.9 ± 14.2	60.9 ± 13.6	68.6 ± 11.9	71.9 ± 10.8	0.0004
Men (%)	69.4	76.1	66.7	69.2	62.5	0.51
Underlying kidney disease						0.72
Diabetic kidney disease (%)	42.8	49.3	35.9	53.8	33.3	
Chronic glomerulonephritis (%)	30.1	26.8	34.6	23.1	41.7	
Nephrosclerosis (%)	20.1	18.3	20.5	15.4	20.8	
Others (%)	7.0	5.6	9.0	7.7	4.2	
HD duration (years)	0.53 (0.51–3.80)	0.53 (0.51–3.62)	0.80 (0.52–4.85)	0.52 (0.51–2.40)	0.53 (0.51–6.15)	0.36
Alcohol (%)	21.0	26.8	17.9	15.4	12.5	0.33
Smoking (%)	25.8	25.4	32.1	23.1	16.7	0.44
Hypertension (%)	96.5	97.2	96.2	96.2	95.8	0.98
Diabetes (%)	45.8	50.7	37.2	65.4	37.5	0.053
History of CVD (%)	62.4	60.6	65.4	65.4	62.5	0.93
Dw (kg)	58.0 ± 12.4	63.0 ± 12.5	58.7 ± 11.0	53.9 ± 10.7	48.5 ± 9.5	<0.0001
BMI (kg/m^2^)	22.2 ± 3.9	23.8 ± 3.5	22.8 ± 3.8	20.7 ± 3.5	18.8 ± 2.7	<0.0001
BUN (mg/dL)	60.3 ± 16.7	61.8 ± 16.2	65.0 ± 17.5	54.2 ± 16.1	59.1 ± 14.6	0.031
Creatinine (mg/dL)	8.9 ± 3.2	9.4 ± 3.2	9.7 ± 3.4	7.2 ± 2.4	8.2 ± 2.1	0.0019
Albumin (g/dL)	3.7 ± 0.4	3.8 ± 0.3	3.9 ± 0.3	3.3 ± 0.3	3.5 ± 0.3	<0.0001
Hemoglobin (g/dL)	10.8 ± 1.3	10.7 ± 1.1	10.9 ± 1.1	11.0 ± 1.1	10.5 ± 1.4	0.27
T-Cho (mg/dL)	154 ± 35	153 ± 39	162 ± 34	156 ± 33	134 ± 18	0.0081
Uric acid (mg/dL)	7.0 ± 1.8	7.1 ± 2.0	7.0 ± 1.8	6.8 ± 1.8	7.1 ± 1.4	0.91
Ca (mg/dL)	8.8 ± 0.9	8.6 ± 0.9	9.1 ± 0.9	8.3 ± 0.6	8.9 ± 1.1	0.0001
Phosphorus (mg/dL)	5.1 ± 1.4	5.3 ± 1.5	5.3 ± 1.3	4.8 ± 1.2	5.1 ± 1.3	0.28
iPTH (pg/mL)	126 (48–219)	158 (59–251)	133 (64–224)	127 (68–206)	89 (19–168)	0.080
Glucose (mg/dL)	139 ± 60	144 ± 69	132 ± 47	139 ± 68	132 ± 56	0.62
C-reactive protein (mg/dL)	0.16 (0.07–0.46)	0.15 (0.07–0.36)	0.10 (0.06–0.23)	0.18 (0.03–0.94)	0.12 (0.06–0.69)	0.55
GNRI at baseline	94.0 ± 7.0	96.8 ± 3.7	98.3 ± 4.1	86.0 ± 4.9	87.0 ± 3.4	<0.0001
GNRI after one year	NA	99.2 ± 3.7	94.0 ± 5.1	90.5 ± 4.4	83.5 ± 4.5	<0.0001
ΔGNRI (%)	NA	1.7 (0.9–3.5)	−3.6 (−5.8 to −2.4)	4.7 (0.4–8.7)	−3.4 (−6.5 to −1.7)	<0.0001

HD, hemodialysis; BMI, body mass index; BUN, blood urea nitrogen; T-Cho, total cholesterol; iPTH, intact parathyroid hormone; CVD, cardiovascular disease; Dw, dry weight; GNRI, geriatric nutritional risk index; ΔGNRI, annual change in GNRI; NA, not applicable. G1 (group1), GNRI ≥ 91.2 and ΔGNRI ≥ 0%; G2, GNRI ≥ 91.2 and ΔGNRI < 0%; G3, GNRI < 91.2 and ΔGNRI ≥ 0%; and G4, GNRI < 91.2 and ΔGNRI < 0%.

**Table 2 nutrients-12-03333-t002:** Cox proportional hazards analysis of baseline GNRI and annual change in GNRI of mortality.

Variables	Unadjusted		Adjusted	
	HR (95% CI)	*p*-Value	HR (95% CI)	*p*-Value
All-cause mortality				
GNRI (continuous)	0.92 (0.89–0.94)	<0.0001	0.94 (0.91–0.98)	0.0033
ΔGNRI (continuous)	0.89 (0.83–0.96)	0.0010	0.89 (0.83–0.95)	0.0006
GNRI < 91.2	3.97 (2.46–6.38)	<0.0001	2.59 (1.54–4.33)	0.0004
ΔGNRI < 0%	2.03 (1.15–3.75)	0.014	2.33 (1.32–4.32)	0.0032
Cross-classified (vs. G1)		0.0003		0.0067
G2	2.11 (1.05–4.60)	0.036	2.68 (1.31–5.93)	0.0061
G3	3.05 (1.03–8.28)	0.045	2.57 (0.84–7.27)	0.095
G4	6.69 (2.87–15.94)	<0.0001	3.88 (1.62–9.48)	0.0026
Cardiovascular mortality				
GNRI (continuous)	0.93 (0.88–0.97)	0.0011	0.94 (0.90–1.00)	0.056
ΔGNRI (continuous)	0.88 (0.80–0.97)	0.0088	0.87 (0.78–0.96)	0.0041
GNRI < 91.2	3.44 (1.68–6.85)	0.0010	2.47 (1.14–5.23)	0.022
ΔGNRI < 0%	2.51 (1.12–6.39)	0.025	3.00 (1.32–7.72)	0.0079
Cross-classified (vs. G1)		0.0093		0.031
G2	2.97 (1.08–10.43)	0.035	3.96 (1.39–14.26)	0.0089
G3	3.78 (0.74–17.32)	0.10	2.98 (0.57–14.07)	0.18
G4	8.44 (2.40–33.12)	0.0012	4.74 (1.29–19.42)	0.020

GNRI, geriatric nutritional risk index; ΔGNRI, annual change in GNRI; HR, hazard ratio; CI, confidence interval; G1 (group 1), GNRI ≥ 91.2 and ΔGNRI ≥ 0%; G2, GNRI ≥ 91.2 and ΔGNRI < 0%; G3, GNRI < 91.2 and ΔGNRI ≥ 0%; and G4, GNRI < 91.2 and ΔGNRI < 0%. All-cause mortality: adjusted for age, creatinine level, and C-reactive protein level. CVD mortality: adjusted for age, history of cardiovascular disease, and creatinine level.

**Table 3 nutrients-12-03333-t003:** Predictive accuracy of annual change in GNRI for mortality.

Variables	C-Index	*p*-Value	NRI	*p*-Value	IDI	*p*-Value
All-cause mortality						
Established risk factors including baseline GNRI	0.702 (0.629–0.775)		Ref.		Ref.	
+ ΔGNRI	0.733 (0.652–0.813)	0.39	0.525	0.0005	0.055	0.0005
Cardiovascular mortality						
Established risk factors including baseline GNRI	0.709 (0.614–0.804)		Ref.		Ref.	
+ ΔGNRI	0.769 (0.666–0.872)	0.15	0.553	0.0034	0.045	0.0034

GNRI, geriatric nutritional risk index; ΔGNRI, annual change in GNRI; NRI, net reclassification improvement; (IDI), integrated discrimination improvement. G1 (group 1), GNRI ≥ 91.2 and ΔGNRI ≥ 0%; G2, GNRI ≥ 91.2 and ΔGNRI < 0%; G3, GNRI < 91.2 and ΔGNRI ≥ 0%; and G4, GNRI < 91.2 and ΔGNRI < 0%.

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
