# Peer review of "Impact of Annual Change in Geriatric Nutritional Risk Index on Mortality in Patients Undergoing Hemodialysis"

_nutrients, 2020, doi:10.3390/nu12113333_

Round 1

Reviewer 1 Report

  1. How relevant are the 11-year all-cause survival rates and 10-year survival rates in a relatively small study with a median follow-up period of 3.7 (1.9–6.9) years? Could this be considered a limitation of the research? Were survival rates for shorter intervals of time calculated? If so, what were the results?
  2. Results for the C-index, net reclassification improvement (NRI), and integrated discrimination improvement (IDI) warrant only a brief (and therefore insufficient) comment in the Discussion section.
  3. The length of the Discussion section is limited and would benefit from more extensive comments.

Author Response

Response to Reviewer 1

Thank you very much for your very constructive comments.

  1. How relevant are the 11-year all-cause survival rates and 10-year survival rates in a relatively small study with a median follow-up period of 3.7 (1.9–6.9) years? Could this be considered a limitation of the research? Were survival rates for shorter intervals of time calculated? If so, what were the results?

Thank you very much for the comments.

We calculated all-cause survival rates for shorter intervals of time. The 5-year all-cause survival rates were 82.1% in higher GNRI group and 43.9% in lower GNRI group, respectively (p <0.0001). The 4-year all-cause survival rates were 88.3% in maintained ΔGNRI group and 70.6% in declined ΔGNRI group, respectively (p = 0.013). The 4-year all-cause survival rates were 91.8%, 79.0%, 79.8%, and 38.4%, in G1, G2, G3, and G4, respectively (p <0.0001). Thus, similar results were obtained. However, we think that many events data were missing if shorten the follow-up period, therefore we would like to show longer follow-up data. However, this is an important limitation. So, we added following sentence, “The 10-year survival data might be unstable because of short median follow-up period.” in the limitation section in the revised manuscript. If reviewer suggest follow-up period should be shorten in the text, of course we will do so. We thank your kind understanding.

  1. Results for the C-index, net reclassification improvement (NRI), and integrated discrimination improvement (IDI) warrant only a brief (and therefore insufficient) comment in the Discussion section.

Thank you very much for the comments.

The C-index for all-cause mortality increased by the addition of ΔGNRI to the established risk model including baseline GNRI from 0.702 to 0.733, but did not reach statistical significance  (p = 0.39). This might be due to small number of patients. However, the addition of ΔGNRI to the predicting model significantly improved the NRI (0.525, p=0.0005) and the IDI (0.055, p=0.0005). The NRI relatively indicates how many patients improve their predicted probabilities, and the IDI also represents the average improvement in predicted probabilities, respectively [14]. On the other words, 52.5% of patients had improved the predictability and the average improved predicted probabilities increased by 0.055 when add ΔGNRI into a predicting model including baseline GNRI. We added these sentences in the discussion section.

  1. The length of the Discussion section is limited and would benefit from more extensive comments.

Thank you very much for the comments. We discussed especially about the model discrimination as you mentioned above.

Reviewer 2 Report

Predicting the future is full of uncertainty. Risk charts can govern treatment options.   The geriatric nutritional risk index (GNRI) was described in 2005 as a simplification of a previous attempt described in the eighties. The index is based on serum albumen and weight. The aim of the study is clearly stated but the last line of the introduction is  less clear. ‘Moreover, we examined whether the change in GNRI can improve the predictability of mortality’ My question is what was the comparison to show improvement? This should be explained. In the introduction the meta analysis by Xiong J et al in 2018 (Kidney blood pressure Res) should be mentioned as the meta analysis concluded that GNRI is associated with, both all cause and CVD mortality in HD patients. The earlier paper in J Nephrol 2014 2 193-201 by Panichi V et al also found that the GNRI index a strong predictor of mortality in HD patients,

Were there only 229 patients undergoing HD for more than 6 months who had maintenance HD between 2008 and 2019 and ? were there  other reasons for non inclusion? I am at a loss to understand the 4 groups as 1 and 2 are the same and 3 and 4 the same. I presume the > and < have got muddled up?  

The results are hard to follow.  Follow up was for 1.9 to 6.9 years and the patients who died in the first year were excluded. So in the 6.9 years how did the baseline GNRI compare to the change in GNRI at the end of the first year. The 10 year survival rates are given but it is difficult to see how much better the change in GNRI as compared to the initial index value. What does the reclassification improvement of 0.525 mean? This should be more clearly stated.  At the end of the first year  the GNRI reclassified how many patients.? Was the change in GNRI a better predictor as compared to the new GNRI.?

The discussion is well written.

In conclusion a well designed, retrospective study carefully carried out. Clarity of the implications of the study findings would help.

Author Response

Response to Reviewer 2

Thank you very much for your very constructive comments.

Predicting the future is full of uncertainty. Risk charts can govern treatment options.   The geriatric nutritional risk index (GNRI) was described in 2005 as a simplification of a previous attempt described in the eighties. The index is based on serum albumen and weight. The aim of the study is clearly stated but the last line of the introduction is less clear. ‘Moreover, we examined whether the change in GNRI can improve the predictability of mortality’ My question is what was the comparison to show improvement? This should be explained.

Thank you very much for the comment. We are sorry for lack of explanation. In the present study, we compared the predictability between in a predicting model with both of baseline GNRI and ΔGNRI vs. a model with baseline GNRI alone. Thus, we revised our manuscript as ‘Moreover, we examined whether the change in GNRI can improve the predictability of mortality when it was added to established risk factors including the baseline GNRI’.

In the introduction the meta analysis by Xiong J et al in 2018 (Kidney blood pressure Res) should be mentioned as the meta analysis concluded that GNRI is associated with, both all cause and CVD mortality in HD patients. The earlier paper in J Nephrol 2014 2 193-201 by Panichi V et al also found that the GNRI index a strong predictor of mortality in HD patients,

Thank you very much for the comments. According to your advice, we cited the paper written by Xiong J et al. as meta-analysis (Reference No. 13). In addition, we also cited the paper written by Panichi V et al (Reference No. 11).

Were there only 229 patients undergoing HD for more than 6 months who had maintenance HD between 2008 and 2019 and ? were there  other reasons for non inclusion?

Thank you very much for the comments. We included all patients who had underwent maintenance hemodialysis for more than 6 months between 2008 and 2019. There were no excluded patients.

I am at a loss to understand the 4 groups as 1 and 2 are the same and 3 and 4 the same. I presume the > and < have got muddled up? 

Thank you very much for the comments. We are sorry for leading your confusion. Patients were divided into four groups according to the baseline GNRI (<91.2 vs. ≥91.2) and declined or maintained GNRI (ΔGNRI <0% vs. ΔGNRI ≥0%) during the first year, namely, G1, higher GNRI and maintained GNRI; G2, higher GNRI and declined GNRI; G3, lower GNRI and maintained GNRI; G4, lower GNRI and declined GNRI.

The results are hard to follow.  Follow up was for 1.9 to 6.9 years and the patients who died in the first year were excluded. So in the 6.9 years how did the baseline GNRI compare to the change in GNRI at the end of the first year. The 10 year survival rates are given but it is difficult to see how much better the change in GNRI as compared to the initial index value.

Thank you very much for the comments. With regards to model discrimination, we did not compare the predictability for mortality of ΔGNRI to that of baseline GNRI. Because, as you mentioned above, GNRI is an established predictor for both all-cause and CVD mortality. Therefore, we evaluated whether the change in GNRI can improve the predictability of mortality when it was added to established risk factors including the baseline GNRI. We thank your kind understanding.

What does the reclassification improvement of 0.525 mean? This should be more clearly stated. 

Thank you very much for the comments. The NRI relatively indicates how many patients improve their predicted probabilities for mortality. On the other words, 52.5% of patients had improved the predictability when add ΔGNRI into a predicting model including baseline GNRI. We added these sentences in the revised manuscript.

At the end of the first year  the GNRI reclassified how many patients.? Was the change in GNRI a better predictor as compared to the new GNRI.?

Thank you very much for the comments. The decline of annual change in GNRI occurred in 78 patients (52.3%) of higher GNRI group (GNRI ≥91.2) and 24 patients (48.0%) of lower GNRI group (GNRI <91.2), respectively. We added these results in our manuscript.

As we mentioned above, we did not compare the predictability for mortality of ΔGNRI to that of baseline GNRI. Because, as you mentioned above, GNRI is an established predictor for both all-cause and CVD mortality. Therefore, we evaluated whether the change in GNRI can improve the predictability of mortality when it was added to established risk factors including the baseline GNRI. We thank your kind understanding.

The discussion is well written.

Thank you very much.

In conclusion a well designed, retrospective study carefully carried out. Clarity of the implications of the study findings would help.

Thank you very much for the comments. As our main findings, the annual change in GNRI (ΔGNRI) could not only predict all-cause and cardiovascular mortality but also improve predictability for mortality. Therefore, GNRI may be proposed to be serially evaluated to precisely predict mortality in hemodialysis patients.

Reviewer 3 Report

This is a well-planned study aimed to investigate if an annual change in geriatric nutritional risk index (ΔGNRI) can predict all-cause and cardiovascular mortality in hemodialyzed patients.

Fortunately, ΔGNRI appeared a good parameter for mortality prediction in these patients. However, it is rather the only novelty of this work. It is widely known that malnutrition is a strong risk factor of death, thus parameters measuring malnutrition or worsening nutritional status also should be.

Although the problem is a kind of interest, the article contains some flaws.

Major comments:

  1. In this work, ΔGNRI was calculated on the base of the annual change of GNRI, whereas 20 patients died in this period. It means that calculating ΔGNRI was completely useless for these deceased patients. In this context, the conclusion that "GNRI should be serially evaluated to precisely predict mortality in this population" is not justified. Probably, 3-months ΔGNRI would be more acceptable.
  2. In results, the Authors stated that "C-index for all-cause mortality improved by adding ΔGNRI to the established risk model including age, creatinine level, C-reactive protein level, and baseline GNRI from 0.702 to 0.733 (p = 0.39)." This is not statistically significant, thus improvement interpretation is not valid.
  3. Models with and without ΔGNRI for all-cause as well as cardiovascular mortality didn't differ in AUC. Moreover, Net Reclassification Index and Integrated Discrimination Index are not appropriate tools for testing prediction improvement of the model because they improperly forced risks and reward overconfidence. Thus, significant improvement in the prediction of the model is questionable.

Author Response

Response to Reviewer 3

Thank you very much for your very constructive comments.

This is a well-planned study aimed to investigate if an annual change in geriatric nutritional risk index (ΔGNRI) can predict all-cause and cardiovascular mortality in hemodialyzed patients.

Fortunately, ΔGNRI appeared a good parameter for mortality prediction in these patients. However, it is rather the only novelty of this work. It is widely known that malnutrition is a strong risk factor of death, thus parameters measuring malnutrition or worsening nutritional status also should be.

Although the problem is a kind of interest, the article contains some flaws.

Major comments:

In this work, ΔGNRI was calculated on the base of the annual change of GNRI, whereas 20 patients died in this period. It means that calculating ΔGNRI was completely useless for these deceased patients. In this context, the conclusion that "GNRI should be serially evaluated to precisely predict mortality in this population" is not justified. Probably, 3-months ΔGNRI would be more acceptable.

Thank you very much for the comments. We fully understand what you mean. We showed that the declined annual change of GNRI was independently associated with all-cause and CVD mortality. In addition, Lee et al. also reported that changes in GNRI during the first year determined risks of major adverse cardiac and cerebrovascular events in peritoneal dialysis patients. Therefore, 1-year change in GNRI may be enough to predict mortality and cardiovascular events. However, as you mentioned above, the optimal duration of changes in GNRI remains unknown. Therefore, this is a limitation of the present study. So, we added the sentence ‘Third, we evaluated the relationships between the annual change in GNRI and mortality, but the optimal duration of changes in GNRI remains unknown.’ in the limitation. Moreover, according to your comments, we corrected the conclusion that “GNRI may be proposed to be serially evaluated to precisely predict mortality in this population”. So, we revised our manuscript thoroughly.

In results, the Authors stated that "C-index for all-cause mortality improved by adding ΔGNRI to the established risk model including age, creatinine level, C-reactive protein level, and baseline GNRI from 0.702 to 0.733 (p = 0.39)." This is not statistically significant, thus improvement interpretation is not valid.

Thank you very much for the comments. We are sorry for our miswriting. We revised our manuscript as followed: C-index for all-cause mortality increased by adding ΔGNRI to the established risk model including age, creatinine level, C-reactive protein level, and baseline GNRI from 0.702 to 0.733 (p = 0.39), but did not reach statistical significance.

Models with and without ΔGNRI for all-cause as well as cardiovascular mortality didn't differ in AUC. Moreover, Net Reclassification Index and Integrated Discrimination Index are not appropriate tools for testing prediction improvement of the model because they improperly forced risks and reward overconfidence. Thus, significant improvement in the prediction of the model is questionable.

Thank you very much for the comment. Pencina MJ, a statistician for Framingham Heart Study, mentioned the utility of reclassification methods such as NRI or IDI to evaluate the added predictive ability of a new marker (we referred in our manuscript as No. 16). In our limited knowledge, these tools have been often used in numerous papers (Zethelius et al, NEJM 358, 2008, Nina et al, Ann Intern Med 150, 2009, etc). Furthermore, we also used these tools in some papers in Nutirents (Ishii et al, 2017, 9, 416; Yajima et al, 2018, 10, 480; Yajima et al, 2019, 11, 2659). We think these tools might be not inappropriate to be used in this study. We thank your kind understanding.

Round 2

Reviewer 1 Report

I read with great interest the current version of the manuscript, and I think the improvements the authors made based on previous reviewer comments served to provide a better value for the text.

Still, I think scientific transparency would require mentioning the all-cause survival rates for shorter intervals of time in the Results section. Perhaps a phrase in the Discussion section, acknowledging the similarity of results with the longer follow-up data, and the risk of missing events the former bring, would also be welcomed.

Author Response

Response to reviewer 1

Thank you very much for your constructive comments.

I read with great interest the current version of the manuscript, and I think the improvements the authors made based on previous reviewer comments served to provide a better value for the text.

Still, I think scientific transparency would require mentioning the all-cause survival rates for shorter intervals of time in the Results section. Perhaps a phrase in the Discussion section, acknowledging the similarity of results with the longer follow-up data, and the risk of missing events the former bring, would also be welcomed.

Thank you very much for the comments.

We added the results of 5-year all-cause survival rate for lower GNRI vs higher GNRI and 4-year all-cause survival rate for declined GNRI vs maintained GNRI and divided four groups in the results section. We also discussed about the similarity of results with the longer follow-up data and the risk of missing events in the discussion section.

Reviewer 3 Report

The authors significantly improved their manuscript. However, the statistical analysis did not change. I maintain my opinion that the Net Reclassification Index and the Integrated Discrimination Index are not appropriate tools for testing the model's prediction improvement because they just force this positive result. (Toxicol Sci. 2017 Mar; 156(1): 11–13; Stat Biosci. 2015 Oct 1; 7(2): 282–295.). Insignificant differences in C-index suggest a lack of model improvement rather. What about the other statistics (R2, AIC, BIC, Harrell's C-index .... )?

Moreover, the Authors should clarify why they put together two significantly correlated variables in one prediction model (GNRI and ΔGNRI)? This multiplies the mutual prediction value of these parameters; thus, the model is improperly enhanced.    

Author Response

Response to reviewer 3

Thank you very much for your constructive comments.

The authors significantly improved their manuscript. However, the statistical analysis did not change. I maintain my opinion that the Net Reclassification Index and the Integrated Discrimination Index are not appropriate tools for testing the model's prediction improvement because they just force this positive result. (Toxicol Sci. 2017 Mar; 156(1): 11–13; Stat Biosci. 2015 Oct 1; 7(2): 282–295.). Insignificant differences in C-index suggest a lack of model improvement rather. What about the other statistics (R2, AIC, BIC, Harrell's C-index .... )?

Thank you very much for the comments. We understand what you mean. Overestimating of NRI considered to be often occurred in poorly calibrated model. In the present study, two model might be well calibrated model (p=0.73 for the baseline GNRI model and p=0.75 for the adding ΔGNRI model by Hosmer-Lemeshow test). Therefore, we think our results might not be completely incorrect. Unfortunately, we do not have fully knowledge of other evaluation tools. Thus, we added a following sentence in the Discussion and Limitation section in the revised manuscript.

“However, recently, some statisticians raised a concern the overestimation of the improvement of predictability among predicting models using the NRI [23,24]. Although the NRI has been used to discriminate predicting model among numerous studies, this point might be paid caution, and our results might have to be conform using other evaluation tools.”

Moreover, the Authors should clarify why they put together two significantly correlated variables in one prediction model (GNRI and ΔGNRI)? This multiplies the mutual prediction value of these parameters; thus, the model is improperly enhanced.    

Thank you very much for the comments.

We calculated variance inflation factor (VIF) in a prediction model including both baseline GNRI and ΔGNRI. VIF for baseline GNRI and that of ΔGNRI were 1.42 and 1.15, respectively. Thus, even though ΔGNRI and baseline GNRI were significantly correlated, we think no worry that the model including both baseline GNRI and ΔGNRI is improperly enhanced.